

# Effects of post-activation protocols based on slow tempo bodyweight squat and isometric activity on vertical jump height enhancement in trained males: a randomized controlled trial

Dawid Koźlenia and Jarosław Domaradzki

Unit of Biostructure, Faculty of Physical Education and Sport, Wroclaw University of Health and Sport Sciences, Wroclaw, Poland

## ABSTRACT

This study aimed to establish the effectiveness of slow tempo bodyweight squat combined with an isometric squat (ST-ISO), and an isometric squat alone (ISO), as a post-activation performance enhancement protocol (PAPE) for jump height improvement. The study sample consisted of 41 trained men aged 18–24.
The ST-ISO group ($n = 17$) performed three five-second sets of the maximal voluntary back squat while pushing on an immovable bar and two sets of five repetitions of a slow-tempo (5-0-5-0) body squat immediately after isometry with a 2-m rest interval. The ISO ($n = 14$) group only performed isometric squats, and the control group (CG; $n = 10$) performed a 5-min treadmill run at 6 km/h.
The countermovement jump (CMJ) height results were analyzed from the baseline and then at 3, 5, 7, and 9 min after the PAPE protocols. The statistical significance was set at $p < 0.05$. RM-ANOVA revealed differences in the group-minute interaction (F = 2.70; $p = 0.0083$; $\eta^2 = 0.1243$), and *post-hoc* tests demonstrated a significant decrease in CMJ after 5 min in the ISO group ($p < 0.0446$). The performance of the ST-ISO group markedly decreased in the 3rd and 7th min after PAPE ($p = 0.0137$; $p = 0.0424$, respectively), though it improved significantly in the final minute ($p < 0.0030$). Chi-squared analysis revealed that the ST-ISO group peaked more frequently in the 9th min ($X^2 = 17.97$; $p = 0.0214$). However, CMJ height improvement did not differ between the PAPE protocols, thus it was close to statistical significance (t = −1.82; $p = 0.07$; ES = 0.7). The ST-ISO protocol provided jump enhancement, though the deterioration observed in the first minutes after the protocols suggest the rest period after activity requires attention, and the methods need to be individualized.

## INTRODUCTION

Jump height is a reliable indicator of athletic performance affected by power output a body mass or force-velocity profile (*Pupo et al., 2020*; *Jiménez-Reyes et al., 2017*). Jump height measurements are standard used in sports control and diagnostics (*Carvalho, Mourão &*

Corresponding author
Dawid Koźlenia,
dawid.kozlenia@awf.wroc.pl

*Abade, 2014*; *Clemente et al., 2022*). It represents the ability to apply force per unit time (RFD) what directly impact athletic performance (*Rodríguez-Rosell et al., 2017*).

The phenomenon called post-activation performance enhancement (PAPE) makes it possible to evoke immediate, temporal improvement in physical performance (*Blazevich & Babault, 2019*). The PAPE effect is caused by an increase body temperature, neural drive improvement, and intracellular fluid migration in response on conditioning activity (*Cormier et al., 2022*). However, fatigue, may suppresses the PAPE effects (*Blazevich & Babault, 2019*; *Prieske et al., 2020*). The critical factors in a PAPE protocol utilization is selecting optimal activity intensity, volume, rest breaks, using an activity with related movement pattern to the targeted effort (*Boullosa, 2021*; *Seitz & Haff, 2016*). Generally, load of 64–80%-repetition maximum (RM) could bring performance improvement in 3–7 min (*Dobbs et al., 2019*). However also, lower intensity could be effective, too (*Boullosa et al., 2018*). Another factor is the type of muscle contraction, with either dynamic or isometric activities providing positive effects, although variability in results is observed (*Rixon, Lamont & Bemben, 2007*).

The literature demonstrates the efficiency of interventions based on isometric efforts with external load (*Vargas-Molina et al., 2021*) and volitional isometry (*Spieszny et al., 2022*). Even though some observations provide data suggested isometry as a superior towards dynamic action, whereas other observed contrary (*Rixon, Lamont & Bemben, 2007*; *Tsolakis et al., 2011*). The efficiency of isometry in evoking a PAPE effect could be associated with improved neural drive and lower metabolic cost than the dynamic effort (*Cady et al., 1989*; *Duchateau & Hainaut, 1984*). Also, combining isometry with other types of muscle contraction may provide positive results (*Kalinowski et al., 2022*). *Kalinowski et al. (2022)* showed improvement in countermovement jump (CMJ) after PAPE protocol based on combination isometry and plyometric activities. It indicates the possibility of combination isometry with other activities.

Typically, in resistance training programming as well in PAPE protocols creation primarily are consider intensity (load), volume and rest break (*Dobbs et al., 2019*; *Wilk, Zajac & Tufano, 2021*). Lastly more attention receive movement tempo which appropriate using may have beneficial effect in increase muscle hypertrophy, strength, and power (*Sakamoto & Sinclair, 2006*; *Newton & Maresh, 2006*; *Headley et al., 2011*) It was proposed by *Wilk, Zajac & Tufano (2021)* to notify four phases of movement: concentric, isometric/transition, eccentric, isometric/transition. Manipulation of these four stages results in overall muscle activation time during the exercise–time under tension. Previous studies showed metabolic response in blood lactate, hormones and muscle activity due to exercise tempo what state a foundation to specific adaptations (*Wilk et al., 2019*, *2020a*, *2020b*; *Hunter, Seelhorst & Snyder, 2003*; *Headley et al., 2011*; *Tanimoto & Ishii, 1985*). Two types of movement tempo during resistance training are indicated: unintentional–which is impacted by the external load and fatigue and intentional–purposely controlled (*Wilk et al., 2018a*; *Wilk, Zajac & Tufano, 2021*). Slower movement tempo increase time under tension and naturally occur when the heavy load is used. Despite this, fast-twitch motor unites are activated when heavy load is used and this type of efforts is also known as effective conditioning activity in performance enhancement (*Blazevich & Babault, 2019*;

*Prieske et al., 2020*; *Boullosa, 2021*; *Seitz & Haff, 2016*). Due to prolonged time under tension slow movement tempo introduced intentionally allow to used lower weights and cause metabolic response (*Usui et al., 2016*; *Wilk et al., 2018a*). This brings a question if intentional slow movement tempo also could provide performance enhancement due to neuromuscular activation. As it was mentioned slow movement tempo did not demand heavy external load therefore bodyweight could be used. *Bampouras & Esformes (2020)* show that full body weight squat protocol can enhance jump height. To date there is lack of data investigated slow movement tempo effect on jump performance or power output. Previously *Wilk et al. (2021)* did not observed decrease in power when the slow tempo was introduced in limited extent in bench press sets.

Using voluntary isometric contractions and slow movement tempo has a low risk of injury due to a low external load and does not require special equipment what also make it is easy to implement in physical effort preparation (*Wilk et al., 2018a*; *Bampouras & Esformes, 2020*; *Toyoshima, Akagi & Nabeshima, 2021*). However, the usefulness of this efforts in acute performance enhancement needs to be analyzed. There is a lack of studies investigating the combination of slow movement tempo and isometry effort in direct preparation for activity. It indicates the area for further investigation. Therefore, this study aimed to establish the effectiveness of PAPE protocols based on slow tempo bodyweight squat combined with volitional isometry (ST-ISO) and volitional isometry alone (ISO) on jump height improvement in trained males and to compare the effects of both protocols. Specifically, we asked: (1) Does the introduced protocols improve jump height compared to the baseline measurement? (2) Which protocols evoke more remarkable improvement? (3) Is there a difference in the time participants take to achieve their peak? (4) Is there a difference in the frequency of positive responses after either intervention? We hypothesize that both introduced protocol provide jump height enhancement. We expect that initially ISO protocol provide better results whereas ST-ISO individuals peaked later due to higher load and more rest will be needed. The obtained results will provide practical information consider PAPE effects using slow tempo and volitional isometry.

## MATERIALS AND METHODS

### Ethics
The Ethics Committee of the Wroclaw University of Sport Health Sciences approved this study (6/2023), which followed the guidelines of the Declaration of Helsinki.

### Study design
This study was a parallel randomized controlled trial. Firstly, volunteers were asked to fill the questionnaire and provide data: age, injury history in last 8 weeks, training experience, use of steroids, and back squat best in last 4 weeks. Then, they were meet on the familiarization session where basic morphological measurements were performed countermovement jump (CMJ) technique was introduced and slow tempo bodyweight squat (5-0-5-0). Then, bar height used for the PAPE protocols was individualized by participants performing a half-squat with a knee angle of 90°. As such, the height of the immovable bar was adjusted based on pushing with maximum effort from a

pre-established position. Also, the back squat strength was verified, the participants were asked to perform the at least one repetition with 120% body weight.

Participants were stratified into one of three groups using the Research Randomizer (www.randomizer.org) tool, including an ST-ISO group, ISO group, and control group (CG). Simple random sampling without replacement was used. The sampling frame was a list of participants in alphabetical order and coded by number 1–56.

During the experimental session participants from all groups completed the same standard warm-up, which involved 5 min of treadmill jogging at 6 km/h, dynamic joint mobilization, lunges, hip thrusts, and two sets of ten body-squat repetitions separated by 60-s brakes. Participants then completed three submaximal CMJs with a 30-s rest between trials. After a 3-min rest, the baseline CMJ was recorded. Participants were asked to perform three as height as possible CMJ with 2 min rest interval. The best result was considered. Then the PAPE protocol was introduced. The ST-ISO protocol included three five-second sets of maximal isometric contraction evoked by pushing an immovable bar with maximum effort, along with two five-repetition sets of full-body squats with 5-0-5-0 movement tempo (Wilk et al., 2018a) controlled by the metronome app (Natural Metronome app, Single Minded Production, LLC, Margate, FL, USA), and with 2-min rest intervals. The ISO group only performed three five-second immovable bar pushes with 1-min rest intervals, and the CG performed a 6-min treadmill run at 6 km/h to maintain preparedness. After PAPE protocols CMJs measurements were completed after 3, 5, 7, and 9 min. The study design is presented in Fig. 1.

## Participants

All participants volunteered, were fully informed about the conditions and risks associated with the study, and provided signed written consent. They could abandon the study at any moment after agreeing to its requirements. Participants had to maintain their daily eating, drinking, and sleeping habits, avoid physical activity for 48 h and refrain from consuming ergogenic compounds or alcohol 24 h before the study.

The initial assessment identified 64 male subjects as potential study participants recruited from members of gym and fitness clubs' members. The inclusion criteria were an age of 18–24 years old, no musculoskeletal injury 8 weeks before the study, no other medical contradictions, experience in strength training of at least 6 years and at least 3 years in continuous resistance training, the ability to perform a back squat with 120% of body weight, whereas exclusion criteria was medical contradiction, using doping substances such as anabolic-androgenic steroids, experience in weightlifting tournaments or actually being professional athlete in any sport discipline, not being able to perform full back squat with 120% bodyweight load. Application of the inclusion criteria led to the exclusion of subjects based on age ($n = 4$) and injury ($n = 4$). No one declare using steroids or experience in weightlifting tournaments. Therefore, in the main stage, 56 subjects were qualified, but one participant from the ST-ISO, four from ISO, and seven from the control group left the study during the experiment and did not complete all measurements— finally, data from 41 subjects (ST-ISO = 17; ISO = 14; CON = 10) were consider. A detailed description of the participants was presented in the section results.

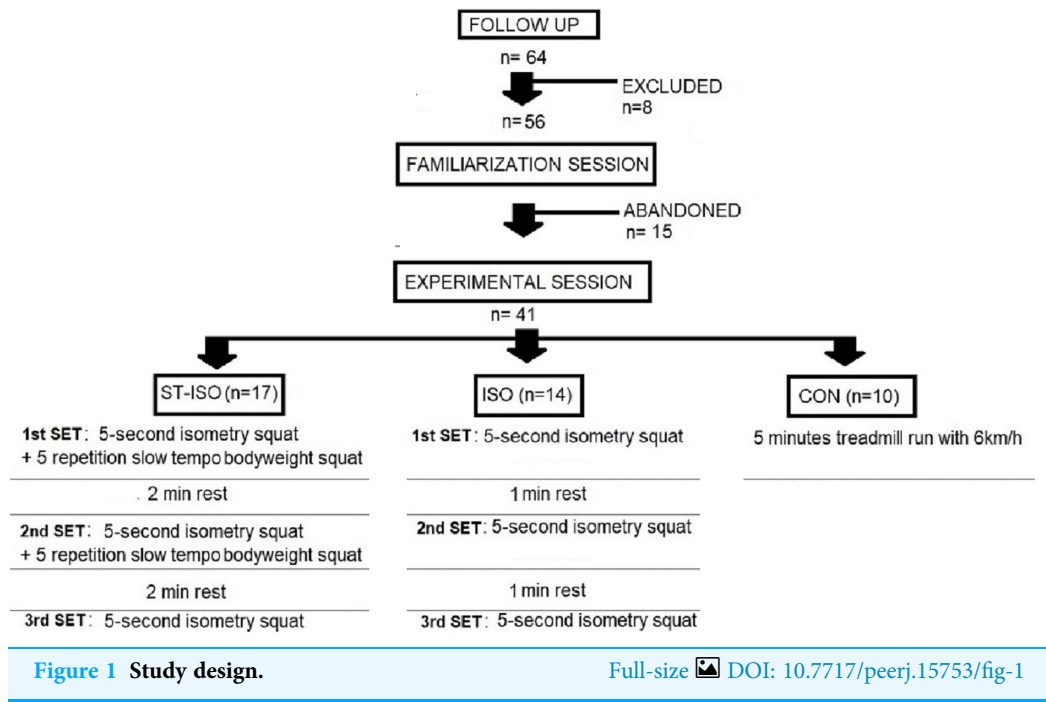

**Figure 1 Study design.**

## Body morphology

Body height was measured at the first meeting during the morning using an anthropometer (GPM Anthropological Instruments, DKSH Ltd., Zürich, Switzerland), and body mass was assessed using an InBody230 device (InBody Co., Ltd., Cerritos, CA, USA), which has been confirmed to be highly reliable (*McLester et al., 2020*).

## Countermovement jump

An OptoJump device (Microgate, Bolzano, Italy) measured CMJ height at a frequency of 1,000 Hz using the equation: $(9.81 \times (\text{flight time})^2/8)$. There were no CMJ depth restrictions, and participants had to keep their arms on their hips and couldn't swing their arms. The validity and reliability of the OptoJump device in jump height measurements were previously confirmed (*Słomka et al., 2017*).

## Statistics

Assuming 95% power and a minimum effect size (ES) of 0.20 at $\alpha = 0.05$, the sample size required to detect changes in the measured parameters was 39 subjects.

The CMJ measurements were validated for test-retest reliability using the intraclass correlation coefficient (ICC 2.1) criteria of *Koo & Li (2016)* was adopted and coefficients of variation (CV), were calculated.

The results were expressed as mean, standard deviation (SD) and 95% confidence intervals (95% CI). The Shapiro–Wilk test examined the data distribution, while Levene's confirmed the homogeneity of variance and Mauchly test was performed to assess data sphericity. A one-way analysis of variance (ANOVA) examined differences in baseline measurements between groups, and a two-way repeated-measures ANOVA (RM-ANOVA) compared differences between groups and time and *post-hoc* test least significant

difference were performed to detect detailed differences between groups. Meanwhile, a chi-squared analysis assessed differences between groups in time-to-achieve peak performance and the frequency of positive protocol responses. Student's t-test for independent samples compared incremental jump height changes between groups. In addition, Cohen's d values for ES were calculated (*Cohen, 1988*). The significance level was set at $p < 0.05$. G*Power was used to determine the required sample size and Statistica 13.0 software (StatSoft Poland, Cracow, Poland) was used for the analysis.

## RESULTS

The ICC 2.1 for CMJ measurements equated to 0.95, and the CV was 0.1, confirming the measurements' reliability.

Table 1 shows descriptive statistics of each group. The one-way ANOVA did not revealed any statistically significant differences between groups in basic parameters. These results indicate a homogenous sample.

In the next of the analysis the jump height results from consecutive minutes provided in Table 2 were compared between three groups.

Two-way RM-ANOVA (group (3) × time (5)) revealed no statistically significant differences for the group (F = 0.16, $p = 0.8484$, $\eta^2 = 0.0085$) though did it for time (F = 4.21, $p = 0.0029$, $\eta^2 = 0.0998$) and group-time interaction (F = 2.70, $p = 0.0083$, $\eta^2 = 0.1243$).

Generally, *post hoc* analysis revealed no significant differences between the protocols. However caution is needed due to quite wide range of the jump height results. Measurements in the ISO group at 5th and 9th min differed significantly ($p = 0.0446$). In this regard, CMJ height decreased in the 5th min and returned to a higher level in the 9th min. In the ST-ISO group, CMJ height decreased in the 3rd ($p = 0.0137$) and 7th ($p = 0.0424$) minutes compared to the baseline. Performance peaked in the 9th min, with an increase observed over the baseline (($p = 0.0030$) 3rd ($p < 0.0001$) 5th ($p = 0.0008$) and 7th ($p < 0.0001$) minutes). In addition, performance decreased to a greater extent in the 3rd min than in the 5th min ($p = 0.0384$) (Fig. 2).

The chi-squared analysis revealed a statistically significant association between the time and individual peak performance ($X^2 = 17.97$, $p = 0.0214$), which, along with the peak performance observed in the 9th min, suggests ST-ISO protocol-related improvement. Also, analysis of the frequency of positive responses to the PAPE protocol revealed that both interventions provided positive reactions compared to the control group (ISO positive responders = 12; 85% *vs* ST-ISO = 15; 88% *vs* CON = 5; 50%, $X^2 = 6.10$, $p = 0.0473$).

In the final analysis, changes in the best results from baseline were comparable between both PAPE protocols. The differences between ISO (0.9 ± 1.2) and ST-ISO (1.3 ± 1.1) were close but insignificant (t = −1.82, $p = 0.0781$), with an ES of 0.7.

## DISCUSSION

This study aimed to describe and compare the efficacy of two PAPE protocols based on volitional maximal isometric effort, with a slow-tempo full-squat added to one of the procedures. Generally, both procedures produced similar effects whereas ST-ISO protocol
**Table 1 Descriptive statistics of the study participants according to the group adherence.** A one-way ANOVA one-dimensional results for all dependent variable.

| Variable | ST-ISO mean (SD) [95% CI] | ISO mean (SD) [95% CI] | CON mean (SD) [95% CI] | F | p | $\eta^2$ |
|---|---|---|---|---|---|---|
| Age (years) | 19.1 (1.5) [18.2–19.9] | 19.9 (1.5) [19.1–20.7] | 20 (1.2) [19.1–20.9] | 1.65 | 0.2062 | 0.08 |
| Body height (m) | 1.8 (0.1) [1.8–1.9] | 1.8 (0.1) [1.8–1.8] | 1.8 (0.1) [1.7–1.8] | 1.30 | 0.2842 | 0.06 |
| Body weight (kg) | 70.5 (13.1) [62.9–78] | 75.7 (9.3) [71–80.5] | 77.6 (4.6) [74.3–80.8] | 1.74 | 0.1894 | 0.08 |
| Training experience (years) | 5.1 (2.1) [3.9–6.4] | 4.4 (1.1) [3.8–4.9] | 4.5 (1.2) [3.7–5.3] | 0.96 | 0.3927 | 0.05 |
| Weekly training volume (min/week) | 277.6 (82.7) [235.1–320.2] | 324.3 (92.9) [270.6–377.9] | 326.0 (58.9) [283.85–368.1] | 1.67 | 0.2003 | 0.05 |

trend to bring more remarkable improvement. However, observation presented some level of uncertainty due elevated range of noted results therefore caution is needed. Despite, deterioration was visible in the first-minute post-intervention, which suggests a requirement for an extended rest period. Indeed, jump height decreased below the baseline in the ST-ISO group, although it improved significantly 1 min after the protocol. Additionally, some ambiguity added jump height fluctuation where decrease and increase occur alternately. The higher decrease during the initial post-protocol period may be related to the higher load associated with the additional slow-tempo squats. On the other hand, the improvement observed after 9 min indicates that the protocol led to appropriate neuromuscular stimulation and evoked a PAPE effect. Moreover, this suggests a relationship between load and rest interval, and an extended break is required when using a higher load.

Using the back squat as a form of conditioning activity to evoke jump height improvement is an effective tool, but avoiding fatigue is crucial (*Krzysztofik et al., 2023*). Therefore, one of the most critical factors associated with utilizing PAPE is an appropriate rest interval, which depends on the load and volume of work. Generally, the break should be approximately 3 min to avoid fatigue and subsequent suppression of the positive PAPE effect (*Dobbs et al., 2019*). Unfortunately, the study protocol, especially with the slow movement tempo added, provided a CMJ height decrease in the initial post-activation minutes. On the other hand, peak performance was visible after 9 min using the ST-ISO protocol, similar to other studies (*Evetovich, Conley & McCawley, 2015*; *Gouvêa et al., 2013*). As such, the type of protocol introduced and individual conditions can strongly affect the results of the PAPE protocol (*Wilson et al., 2013*). Many studies have demonstrated the effects of relative external load (*Đurović et al., 2022*; *Villalon-Gasch et al., 2022*), and others showed peak performance after 8–12 min. Some individuals require an extended rest period (*Gouvêa et al., 2013*; *Kilduff et al., 2008*).
**Table 2 Countermovement jump (CMJ) results according to group adherence measured in consecutive minutes.**

| Variable | ST-ISO mean (SD) [95% CI] | ISO mean (SD) [95% CI] | CON mean (SD) [95% CI] |
|---|---|---|---|
| CMJ baseline (cm) | 37 (4.6) [34.3–39.7] | 38 (6.4) [34.7–41.3] | 37.5 (2.9) [35.4–39.6] |
| CMJ 3′ | 36.5 (4.9) [33.7–39.3] | 36.9 (6.3) [33.7–40.2] | 37.3(3.6) [34.7–39.9] |
| CMJ 5′ | 36.2 (5.1) [33.3–39.2] | 37.8 (6.1) [34.7–41] | 37.4 (3.5) [34.9–39.9] |
| CMJ 7′ | 36.9 (5.2) [33.9–39.9] | 37.1 (6.1) [34–40.3] | 36.7 (4.1) [33.8–39.7] |
| CMJ 9′ | 37.2 (5.3) [34.1–40.2] | 39.3 (6.3) [36–42.5] | 37.1 (3.7) [34.5–39.7] |
| CMJ best | 38 (5) [35.1–40.8] | 39.7 (6.4) [36.5 to −43] | 37.9 (3.5) [35.4–40.3] |
| CMJ best-baseline | 0.9 (1.2) [0.2–1.7] | 1.7 (1.2) [1.1–2.3] | 0.3 (1) [−0.4 to 1] |

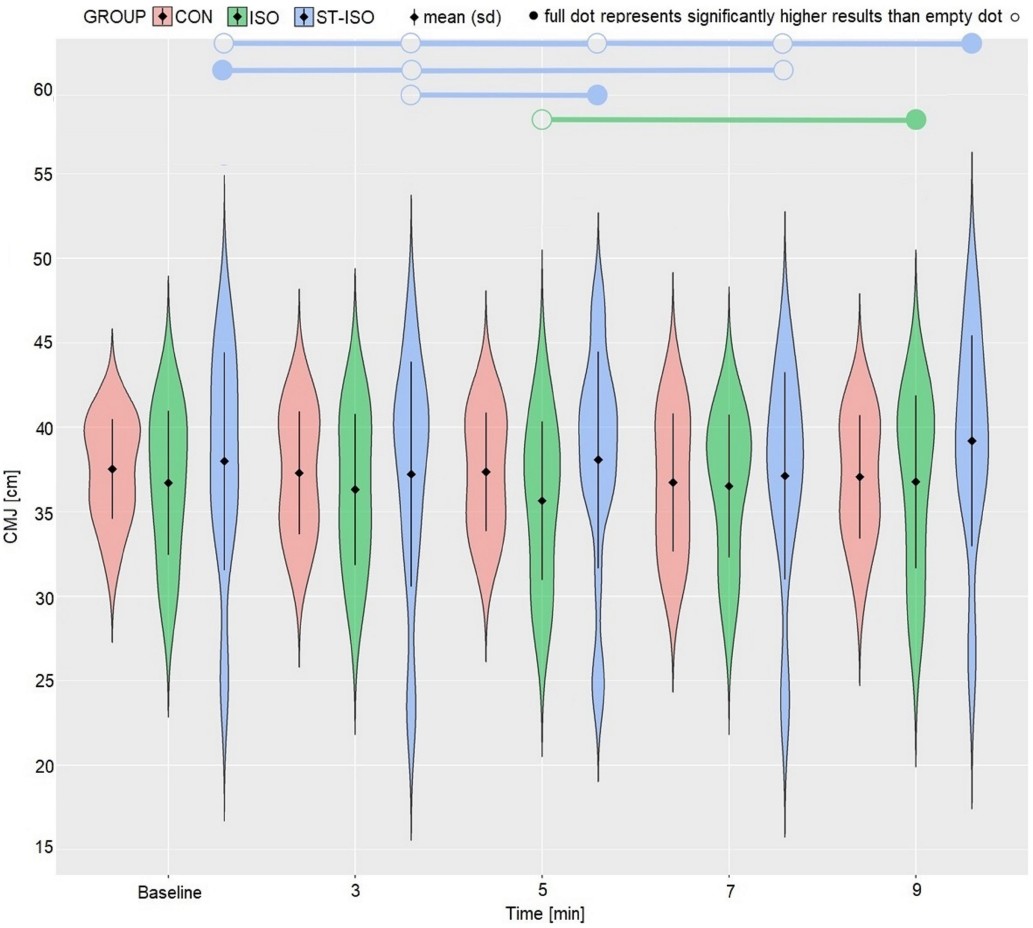

**Figure 2 Changes in jump height over the time of measurements according to group adherence.**

Using isometric effort-based PAPE protocols effectively evokes athletic performance improvement (*Vargas-Molina et al., 2021*; *Hirayama, 2014*). Some observations even suggest that it could be superior to other types of muscle contraction (*Rixon, Lamont &*
*Bemben, 2007*), although many other factors are generally decisive in PAPE protocol efficiency (*Boullosa, 2021*). Indeed, isometric contractions can utilize an external load (*Vargas-Molina et al., 2021*) or voluntary contractions (*Spieszny et al., 2022*), and both activities seem to evoke jump height improvement effectively. *Spieszny et al. (2022)* improved jump height in male team sports players after isometric contraction (pushing on an immovable bar). The peak performance occurred in some individuals after 4 min, whereas others peaked after 8 min, highlighting the importance of the particular conditions on the PAPE response. However, the current study did not show a similar effect. Evoking volitional maximal effort depends strongly on the volition, motivation and personality of individuals (*Ikai & Steinhaus, 1961*; *Welch & Tschampl, 2012*; *McNair et al., 1996*).

Isometry has the potential to be combined with other types of activity, with *Kalinowski et al. (2022)* showing jump height improvement after isometric contractions alongside plyometrics. Other way, *Bampouras & Esformes (2020)* demonstrated jump height improvement evoked by bodyweight squats when using the appropriate volume of activity. In our study we also utilized isometry activity and combined it with bodyweight squat used the slow-tempo (5-0-5-0) proposed by *Wilk et al. (2018a)*. Multiple studies showed that duration of approximately 5s in concentric/eccentric phase led to prolonged time under tension (*Hortobágyi et al., 1996*; *Wilk et al., 2018b*; *Wilk, Tufano & Zajac, 2020*; *Wilk, Zajac & Tufano, 2021*). Limited data exist on the acute effects of slow tempo on dynamic activity as jump performance, though utilizing this tool in training may lead to beneficial long-term training adaptation (*Wilk, Tufano & Zajac, 2020*). Our study showed an immediate decrease in CMJ height in the initial minutes after the PAPE protocol, although more individuals peaked after 9 min and improved their results. Despite, *Goto et al. (2008)* found elevation in catecholamines after slow movement tempo exercise without changes in jump performance in few hours' observation. Our finding suggests that ST-ISO leads to acute fatigue and then evokes activation that leads to CMJ improvement (*Tran, Docherty & Behm, 2006*). Slow movement tempo results in prolonged time under tension and despite lack of external load long-time muscle activity led to higher muscle fibers activity which are crucial in strength/power efforts (*Burd et al., 2012*; *Wilk et al., 2020a*; *Schoenfeld, Ogborn & Krieger, 2015*; *Schuenke et al., 2012*; *Wilk et al., 2021*) therefore, after initial fatigue, and when the recovery occur neuromuscular activity was still present, and the jump enhancement occur. The obtained data firmly indicate setting an optimal rest interval before the targeted effort. There is need to investigate the patterns of motor unit recruitment in response to slow movement tempo to elucidate the association with performance enhancement.

The study was limited by not using a force platform or dynamometer to control the force output of the participants, both of which inform on the stimulus magnitude during voluntary isometry. It also made enable to directly measures power and force during the jump measures and analyzed these parameters. Also, post-activation CMJ height measurements did not extend beyond the time of peak performance in the ST-ISO group. Only males' sample was examined, therefore is need to investigate the effects in females. Future research should address these limitations.

## CONCLUSIONS

Both protocols led to a deterioration in performance in the initial post-activation minutes, with a more remarkable decrease in the ST-ISO group and a subtler decrease in the ISO group. However, the ST-ISO group improved in the following minutes and peaked at the final measurement time, while the ISO group returned to baseline levels. Nonetheless, decreased CMJ height magnitude indicated that ST-ISO increased fatigue more than ISO. The significant improvements recorded in the minutes following this decrease suggest that prolonging the rest period by at least 9 min should achieve better results. However, when compare absolute difference of improvement between both protocol ST-ISO trend to be more effective in jump height enhancement than ISO. The results of this study revealed some positive effects of combining slow movement bodyweight squats with isometric activity. However, caution is demanded when interpreting these results due to some ambiguities associated with the observed alternating pattern of jump height decrease and increase, also observed results are characterized by quite wide range. Therefore, further studies are necessary to extend the observation time and include a larger sample size, which could allow for explaining the observed differences in more detail.

### Funding

This study was supported by the Wroclaw University of Health and Sport Sciences. Grant number Z.22-10. The funders had no role in study design, data collection and analysis, decision to publish, or preparation of the manuscript.

### Grant Disclosures

The following grant information was disclosed by the authors:
Wroclaw University of Health and Sport Sciences: Z.22-10.

### Competing Interests

The authors declare that they have no competing interests.

### Author Contributions

- Dawid Koźlenia conceived and designed the experiments, performed the experiments, analyzed the data, prepared figures and/or tables, authored or reviewed drafts of the article, and approved the final draft.
- Jarosław Domaradzki analyzed the data, authored or reviewed drafts of the article, and approved the final draft.

### Human Ethics

The following information was supplied relating to ethical approvals (*i.e.*, approving body and any reference numbers):

This study was conducted according to the guidelines of the Declaration of Helsinki, and the Ethics Committee of the Wroclaw University of Sport Health Sciences approved this study (6/2023).
### Ethics

The following information was supplied relating to ethical approvals (*i.e.*, approving body and any reference numbers):

The Ethics Committee of the Wroclaw University of Sport Health Sciences approved the study (6/2018).

### Data Availability

The raw measurements are available in the Supplemental Files.

### Supplemental Information

Supplemental information for this article can be found online at http://dx.doi.org/10.7717/peerj.15753#supplemental-information.

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
