# Peer review of "Effects of post-activation protocols based on slow tempo bodyweight squat and isometric activity on vertical jump height enhancement in trained males: a randomized controlled trial"

_PeerJ, doi:10.7717/peerj.15753_

## Round 0.1 · original submission · Major Revisions

Dear authors,

Expert reviewers have provided very detailed and useful comments. A lot of work remains to be done before the paper can be accepted. I look forward to receiving a revised version of the manuscript if the authors believe they can substantially improve the paper (improve readability, provide a clear rationale for the study, more details regarding the methodology, etc.).

Kind regards,
Amador

Reviewer 1 ·

Basic reporting

This study aimed to evaluate the effectiveness of slow tempo bodyweight squat combined with an isometric squat (ST-ISO) and an isometric squat alone (ISO) as a post-activation performance enhancement protocol (PAPE) for improving jump height. The study included 41 trained men aged 18-24 who were divided into three groups: ST-ISO, ISO, and a control group. The countermovement jump height was analyzed at baseline and then at various intervals after the PAPE protocols. The results showed that both PAPE protocols provided jump enhancement, but the rest period after activity needs attention, and the methods need to be individualized. However, there was no significant difference between the two PAPE protocols regarding CMJ height improvement.
Overall, the introduction provides a clear and concise overview of the topic of post-activation potentiation (PAP) and its potential benefits for athletic performance. The introduction presents the main research questions and hypotheses, and it highlights the significance of the study for understanding the underlying mechanisms of PAP and for informing practical training recommendations.
However, there are a few areas where the introduction could be improved. Firstly, the literature review could be more comprehensive and up-to-date. Although the authors cite some relevant studies, there are several key studies that are missing from the review, which could weaken the argument for the study's significance. Additionally, the introduction would benefit from more explicit statements about the specific aims and objectives of the study, as well as more details about the study design and methods. This would help readers to better understand the context and rationale of the study.
Finally, the introduction could benefit from more attention to style and grammar. There are some awkward phrases and sentence constructions that could be revised for clarity and readability. Additionally, there are a few typos and punctuation errors that should be corrected. Overall, the introduction has the potential to be a strong foundation for the study, but some revisions and additions would be needed to make it more robust and compelling.

Experimental design

I would like to commend the authors for providing a clear and concise description of their experimental design. The study was approved by the Ethics Committee and followed the guidelines of the Declaration of Helsinki, which demonstrates a high level of ethical conduct.
However, I would like to point out some areas that could be improved. Firstly, the authors should provide more information on how the participants were recruited and selected for the study. It would also be helpful to include information on the inclusion and exclusion criteria used in the study.
Secondly, while the study design is adequately described, there are some ambiguities that need to be addressed. For example, it is unclear how the participants were randomized into the three groups using the randomizer.org tool. The authors should provide more details on the randomization process to ensure that the groups were truly comparable at baseline.
Thirdly, the authors should provide more information on the training status of the participants. Were they all trained athletes or did the study include individuals with varying levels of training experience? This information is crucial as it could affect the study outcomes.
Lastly, the authors should provide more information on how the data were analyzed. While it is mentioned that the CMJ trials were recorded at various time points after the PAPE protocols, it is unclear how the data were analyzed to determine the effect of the different protocols on jump performance.
Overall, this statistical section has several significant flaws that need to be addressed before publication.
Firstly, the authors claimed that the sample size required to detect changes in the measured parameters was 44 subjects, based on a power analysis using GPower version 3.1.9.6 software. However, the reviewer has run the same software and did not obtain the same sample size of 44. The authors need to explain why their calculation differs from the reviewer's calculation.
Secondly, the organization of the statistical section is confusing. The authors jump between discussing the software used, the statistical tests performed, and the criteria for validating the measurements. This lack of clear structure makes it difficult to follow the authors' argument. The authors should reorganize this section to present the information in a more logical and coherent way.
Thirdly, the authors performed several statistical tests, including ANOVA, RM-ANOVA, chi-squared, and t-test. While some of these tests are appropriate for both parametric and non-parametric data, others are only suitable for either parametric or non-parametric data. It is not clear why the authors chose to use all these tests, and this seems like an excessive amount of statistical tests. The authors need to explain why they used each test and justify their choices. Additionally, the authors should consider whether there is a more parsimonious way to analyze the data, rather than using multiple tests.

Validity of the findings

The study aimed to compare the effects of two protocols of post-activation potentiation exercise (PAPE) on countermovement jump (CMJ) height. The authors concluded that both interventions provided positive reactions compared to the control group, with no significant differences between the protocols.
However, there are some issues that need to be addressed. Firstly, Table 1 only presents descriptive statistics of each group, which do not include any statistical tests performed. This table needs to be modified to include the appropriate statistical tests performed on the data.
Secondly, Figure 1 takes up a lot of space and does not add significant value to the manuscript. The authors should remove it and explain the results in words instead.
Thirdly, Figure 2 is not appropriate for presenting the data. The vertical axis of the graph starts at 33 and ends at 44 cm without any justification. Additionally, the horizontal axis is labeled with the unit of the variable (Minute) instead of min, which is the correct abbreviation according to international standards. It should be labelled as VARIABLE (min). There are no error bars, standard deviation, or confidence intervals, making it difficult to interpret the data. The authors should consider presenting a scatter plot with bar charts to show the changes in jump height over time, along with arrows indicating statistical significance. Alternatively, they could use a box and violin graph to show the distribution of the data.
Lastly, the authors' conclusion that there were no significant differences between the protocols should be taken with caution. Although the differences in the best results from baseline were not significant, there was a trend towards significance with an effect size of 0.7. Therefore, the authors should provide a more nuanced interpretation of their results.

Additional comments

Overall, the study provides some valuable insights into the efficacy of two PAPE protocols based on volitional maximal isometric effort, but there are several limitations and methodological flaws that should be addressed in future research.
Firstly, the study did not use a force platform or dynamometer to control the force output of the participants during voluntary isometry, which could have provided more accurate data on the stimulus magnitude. This could be seen as a significant limitation of the study, as it could affect the validity of the results and their interpretation.
Secondly, the post-activation CMJ height measurements did not extend beyond the time of peak performance in the ST-ISO group. This is another limitation of the study, as it may not provide a comprehensive picture of the effects of the protocols on jump height over a longer period.
Thirdly, the study protocol did not provide a clear justification for the selection of the rest interval used in the study, which was three minutes. This may be an issue, as there is evidence to suggest that the appropriate rest interval for PAPE protocols may vary depending on the load and individual conditions.
Fourthly, the study protocol did not provide a clear explanation of how the slow-tempo (5-0-5-0) proposed by Wilk et al. in bodyweight squats combined with voluntary isometry was expected to lead to improved jump height. This lack of detail could be seen as a limitation of the study, as it may make it difficult to replicate the protocol or interpret the results.

·

Basic reporting

• Good use of English, professional and clear at all times
• Up-to-date, sufficiently cited references also provide an important context of the field of study and its discussion.
• Figures should be enhanced by better quality ones, Figure 1 should frame the flowchart and Figure Two should indicate with asterisks when the differences were statistically significant; Table 1 can be improved by editing the presentation of means, standard deviations and confidence intervals, as well as adding a table footer explaining the variables; And a table 2 should be created presenting the main results, and not just leaving them as text.
• The research questions were addressed correctly, however, in the introduction it is not made clear what the hypotheses are, which were initially had.

Experimental design

• It is novel research, which will help fill a gap in knowledge, it would be important to know in the introduction the biological plausibility on which they relied to determine the use of squats with their own body weight for PAPE.
• Well-defined research questions also denote the relevance of the study and its potential.
• The research is carried out ethically, the rigor could be greater using instruments that allowed to evaluate the force used in the protocol to quantify the load accurately.
• Methods very well explained, with sufficient detail.

Validity of the findings

• It is an interesting topic, since this type of stimulus has not been investigated as PAPE and the results show very promising effects, however it is important to know how the intensity was quantified at the time the slow tempo was performed.
• A correct statistical analysis was carried out, and the data are precise for what you want to answer.
• Well-defined conclusions

Additional comments

General corrections

Format:

• Remember that the citation format in the journal is:
Journal reference format: List of authors (with initials). Year of publication. Full title of the article. Full title of the Journal, volume: page length. DOI (if available).
So, the parenthesis of the year is left over in all citations.


Abstract:
• In keywords change the word "slow tempo" to a Mesh term.

Introduction
• Line 43: Although in this quote it is said that the jump height is an indicator of power, this today can be questioned, since the height of the jump also depends on the imbalance of force, speed and the thrust distance as mentioned (Morin and Samozino in 2017), but I would recommend indicating how González-Badillo and Rodriguez-Rosell show it. (2019) that jump height is an indicator of the ability to apply force per unit time (RFD), and therefore be an indicator of sports performance.
• Line 63-70: Here it would be worth mentioning why use slow tempo without loads, since there it is argued only about the lack of clarity of protocols with slow tempos, but it would be important to mention the biological plausibility of this, perhaps citing works in which physiological changes are demonstrated with the use of slow tempos in self-loading exercises.
• Lines 79-84: mention after the questions, the hypotheses that were had in this regard.

Material and methods:
• Line 108: was only one reference CMJ taken? or were several attempts made and the best, or average, was taken to clarify this situation, since taking only one jump may not be very reliable.
• Figure 1: Make the concept map with lines and boxes, since this looks somewhat complex
• Line 139: raise the number two of the formula (since that two without raising can be confusing) and use Word formula formatting.
• Line 152: it is recommended to present the data as Mean (SD) [95% IC] as indicated by Altman, D. G., & Bland, J. M. (2005) in the text entitled "Standard deviations and standard error", where they indicate:
"In many publications a ± sign is used to join the standard deviation (SD) or standard error (SE) to an observed mean—for example, 69.4±9.3 kg. That notation gives no indication whether the second figure is the standard deviation or the standard error (or indeed something else). A review of 88 articles published in 2002 found that 12 (14%) failed to identify which measure of dispersion was reported (and three failed to report any measure of variability). The policy of the BMJ and many other journals is to remove ± signs and request authors to indicate clearly whether the standard deviation or standard error is being quoted. All journals should follow this practice."

Results
• Table 1: It is recommended to present data as Mean (SD) [95% IC] as mentioned above.
• Lines 173—188: I would recommend placing all results in a "table 2" where the results are more visual. There they place the means with the IC, significance and ES.
• Figure 2: Mark with asterisks the moments where the differences were significant.

Discussion
Lines 229-230: This statement leads us to recommend future studies in which external loads are used for slow tempo.

Reviewer 3 ·

Basic reporting

The manuscript is not well written with regards to language (some parts are very hard to follow for the reader), it may need some major improvements throughout. Although the topic is interesting, this paper showed major concerns with regards to the introduction, hypothesis, methods and design, procedures, discussion and conclusion. The latter raises serious concerns when analyzing results and discussion/conclusions derived. In fact, while the aim should be more clear, the variables are not clearly explained to introduce the general reader into the problem. Most importantly, there is no detailed description of the procedures which makes very hard to understand the tests performed or replicate the study. With only one simple metric reported, is very hard to follow or either understand authors conclusion and why other metrics were not considered.

Experimental design

Testing procedures are not well described, with CMJ height being the only isolated metric reported.
Overall, the study lacks rigorousness, and the methods are not described with sufficient detail & information to be replicated.

Validity of the findings

In summary, my major concerns are:
1. Language
2. Introduction, the rationale for conducting the study is not clear while the gaps in the literature have not been identified/addressed.
3. Testing procedures not well described, with CMJ height as the only isolated metric reported.

Additional comments

Specific Comments
Abstract Section
Lines 18-19: it should be isometric SQ followed by bodyweight squat slow tempo. What is the scientific rationale of using this combination? if the aim is to improve CMJ performance, a more 'explosive' approach should have been used here.

Introduction
I strongly recommend authors to seek for assistance with regards to language. Wording of the first sentence needs revision.
Lines 41-44: the entire paragraph need to be revised, as all of the sentences/ideas are not properly linked between each other.
Line 46: is it power output the right metric? or is it better to overall mention that PAPE may actually improve performance. The sentence need to be reworded. Additionally, PAPE aim is not to increase body temperature.... these are the effects provoked by the exercises and protocols used.
I suggest authors to read the following study:
Cormier, P., Freitas, T. T., Loturco, I., Turner, A., Virgile, A., Haff, G. G., ... & Bishop, C. (2022). Within session exercise sequencing during programming for complex training: historical perspectives, terminology, and training considerations. Sports Medicine, 52(10), 2371-2389.
Lines 54-62: so far, there is a lack of connection on the rationale of the study. The wording also needs to be revised, as it is hard to follow.
Lines 63-64: Which data are the authors referring to? And why is slow-tempo needed?
Lines 67-68: But the key... please revise wording, and how this relates/connects to previous sentence.
Line 71: authors reported above that 'longer' tempo on the eccentric phase may lead to a reduced strength and power, hence, this line don’t make sense.
Additionally, authors mention that isometric contractions and slow tempo have a low risk of injury due to low external loads. Well, why are isometrics and slow tempos implemented with low loads? and no equipment needed? this is not right, please review the entire paragraph.
Lines 75-84: this should be stated in a separate paragraph. Additionally, the questions provided here are not needed and are outside the scope of the study (given the aims provided).
The section lacks a clear rationale, with the protocols used no justified and no hypothesis stated. These are all major issues to be addressed by the authors.
The article doesn’t include sufficient introduction and background to demonstrate how the work fits into the broader field of knowledge. The study, also lacks of relevant prior literature that has analysed this specific topic.
Methods Section
There are several parts in the section that need to be revised regarding language and wording.
Make a more reader friendly figure to illustrate the procedures.
Participants characteristics, please provide more information regarding training experience, resistance training exposure and frequency, etc.
Lines 124-125: how was the 120% BM rule checked? did authors tested the squat prior to inclusion?
Lines 135-136: why reporting BMI? not needed and highly questionable.
Why not reporting other additional metrics than just jump height? additional jump metrics may have provided a greater understanding around the results.

---

## Round 0.2 · Minor Revisions

Some changes required to be done before the article can be accepted for publication.

Reviewer 1 ·

Basic reporting

Dear authors,

I have reviewed the revised manuscript and I am pleased to see that you have addressed all of the concerns raised in my previous review. However, there are still some points that need to be addressed before the manuscript can be accepted for publication.

Please carefully consider these points and make the necessary revisions:

In my initial review, I requested improvements to Figure 2. Although the authors have added 95% confidence limits, there are still some issues. The vertical axis still lacks clear start and end values, while the horizontal axis is not properly labeled, as "minute" is not a variable. Additionally, the use of an apostrophe as a symbol for minute needs clarification. Furthermore, the 95% confidence intervals overlap to such an extent that the high uncertainty makes it difficult to observe any changes between groups (ST-ISO vs. ISO vs. CON at each time point) or within groups (ST-ISO, ISO, CON over the four time points). Surprisingly, the paper's discussion fails to address any of these issues, and the results have been deemed acceptable despite the high degree of uncertainty.

Experimental design

No further comments.

Validity of the findings

No further comments.

Additional comments

No further comments.

---

## Round 0.3 · Minor Revisions

Please, consider the changes proposed by the reviewer for improving the quality of the figures.

Reviewer 1 ·

Basic reporting

No comments, everything was already discussed, thank you.

Experimental design

No comments, everything was already discussed, thank you.

Validity of the findings

No comments, everything was already discussed, thank you.

Additional comments

Dear Authors,
In my previous review, I stated that the vertical axis still lacked clear start and end values, and that the 95% confidence intervals overlapped to such an extent that the high uncertainty makes it difficult to observe any changes between groups. The solution was to change the graph into a violin graph, but the first issue still holds: the first label of the vertical axis is 20 cm but one of the violin start points is located somewhere below 20 cm which I am not able to decipher. The same applies to the upper end of the vertical axes. Please label start and end so that readers can unambiguously know the data values. Finally, I don't think a typeface of around 8 pt would be seen correctly when this figure will be fitted into the pdf. I suggest increasing the size of the font.

---

## Round 0.4 · accepted · Accept

Authors have properly addressed the reviewers´comments.